# Large-angle Lorentz Four-dimensional scanning transmission electron microscopy for simultaneous local magnetization, strain and structure mapping

Sangjun Kang [1,2,3], Maximilian Töllner[1], Di Wang [1,3], Christian Minnert [4,5], Karsten Durst[4], Arnaud Caron[6,7], Rafal E. Dunin-Borkowski [8], Jeffrey McCord [9,10], Christian Kübel [1,2,3] ✉ & Xiaoke Mu[1,11] ✉

Small adjustments in atomic configurations can significantly impact the magnetic properties of matter. Strain, for instance, can alter magnetic anisotropy and enable fine-tuning of magnetism. However, the effects of these changes on nanoscale magnetism remain largely unexplored. In particular, when strain fluctuates at the nanoscale, directly linking structural changes with magnetic behavior poses a substantial challenge. Here, we develop an approach, LA-Ltz-4D-STEM, to map structural information and magnetic fields simultaneously at the nanoscale. This approach opens avenues for an in-depth study of structure-property correlations of magnetic materials at the nanoscale. We applied LA-Ltz-4D-STEM to image strain, atomic packing, and magnetic fields simultaneously in a deformed amorphous ferromagnet with complex strain variations at the nanoscale. An anomalous magnetic configuration near shear bands, which reside in a magnetostatically high-energy state, was observed. By performing pixel-to-pixel correlation of the different physical quantities across a large field of view, a critical aspect for investigating industrial ferromagnetic materials, the magnetic moments were classified into two distinct groups: one influenced by magnetoelastic coupling and the other oriented by competition with magnetostatic energy.

Atomic structures including interatomic distances, coordination, and chemical species play a pivotal role in determining the magnetic properties of matter[1–7]. Characterizing this structure and correlating it with the magnetization state at the nanoscale is essential for unraveling the fundamentals of magnetism and advancing the development of magnetic materials with improved performance[4,8–17]. For instance, atomic strain is a key parameter to manipulate material anisotropy leading to anomalous magnetic properties, e.g., magnetic hardening was observed in a uniaxially strained Ni film[9] or small tensile strains can induce significant anisotropic deformation of magnetic skyrmions in chiral FeGe magnets[18]. These effects enable a broad array of applications, including magnetostrictive devices, magnetic sensors, memory storage, and actuators[9,11,19].

Despite these advances in applications, the correlation between nanoscale strain and magnetism remains poorly understood. This is primarily because current characterization methods cannot provide direct nanoscale correlation of the magnetic and structural information at the same time. Most previous studies on strain-induced magnetic phenomena at the nanoscale have focused on uniaxial and simple strain conditions[9,18]. However, cases where strain varies locally at the

nanometer scale, both common and significant in nanomagnetism, have not been explored quantitatively.

Amorphous ferromagnetic alloys have attracted great interest owing to their magnetic softness together with substantial magnetostriction, contributing to the field of ultrasensitive magnetic field sensors[1,20,21]. Their magnetic properties can be intrinsically altered by straining, in particular, through shear banding during plastic deformation which generates complex strain fields around shear bands[22], altering local magnetic configurations[15]. Until now, these characteristic structures have been studied independently without correlation, primarily due to the experimental challenges. As a result, the impact of deformation on magnetic properties remains poorly understood, hindering further development and application of this material.

With advances in electron microscopy, conventional Lorentz scanning/transmission electron microscopy (Ltz-S/TEM) can now routinely capture magnetic fields at the nanoscale within a sample[9,23–25]. The Lorentz mode, available for both TEM and STEM configurations, provides magnetic field-free conditions at the sample position by eliminating the use of the objective lens that produces strong magnetic fields and may alter the magnetization state of the sample. However, operating commercially available microscopes in this mode using either Fresnel contrast, electron holographic phase information, or STEM-based differential phase contrast (DPC) comes at the expense of spatial resolution and lacks structural information due to the enlarged spherical aberration of the optical system even with aberration correction[26–29]. Recently atomic imaging in field-free conditions has been demonstrated in a specially designed microscope with a developed objective lens hardware, though it is currently difficult for wide access[30]. Moreover, direct real-space atomic imaging limits the field of view to a few tens of nanometers, far less than the typical length scale of magnetic domains or domain walls, which can be hundreds of nanometers or larger, especially in soft magnetic materials. Unlike structural investigation relying on high-resolution atomic lattice imaging, in principle, four dimensional (4D)-STEM can map atomic structures over a large field of view by recording information in reciprocal space while scanning a nanoscale electron probe across the sample[22,31,32]. However, diffracted beams containing structural information are blocked by the column liner tube in field-free STEM mode in all commercially available microscopes. Only the direct beam can be used for the imaging of magnetic domains[33,34]. Direct correlation of the local atomic strain and magnetization is thus difficult.

Here, we advanced field-free optics for STEM, by customizing the configuration of the existing lenses in a standard microscope, to access large diffraction angles in Ltz-STEM mode. The approach can be adopted for all kinds of commercial TEMs equipped with either a Lorentz lens or an image-aberration corrector. Based on the optical configuration, we developed a technique to simultaneously map structural information and magnetic fields using 4D-STEM, which we refer to as large-angle Lorentz 4D-STEM (LA-Ltz-4D-STEM). The method enables simultaneous recording of both the unscattered electron beam and diffracted beams, which provides at the same time the phase shift of the electron exit wave induced by the local magnetic field and the information on the interatomic distance within the sample. We applied the method to directly visualize the local magnetization and strain interaction in a shear band region of a deformed Fe-based amorphous alloy. Using pixel-level correlation between the magnetic and strain field maps, versatile analyses of the mutual interaction between the atomic structure and the local magnetization in the material were achieved.

## Results and discussion

Figure 1 schematically shows the LA-Ltz-4D-STEM idea for simultaneous local magnetization, strain, and structure mapping. The optical setups for nano-beam diffraction, conventional Ltz-STEM, and LA-Ltz-4D-STEM are compared in Fig. 1a–c. The pole pieces of the objective lens are turned off in the Ltz-STEM mode (used in conventional microscopes, Fig. 1b). The diffraction lens, which is used to collect the electrons scattered from the sample and creates a diffraction pattern, is located far away from the sample (more than 10 cm for uncorrected systems). The part of the liner tube between the sample and the diffraction lens acts as a physical aperture and blocks diffracted beams at high angles, thus hindering the recording of atomic-level structural information (the recording angle is limited to <2.2 mrad, corresponding to observable lattice spacings larger than ~0.9 nm on a Themis Z). In order to overcome this issue and to get access to highly scattered electrons containing structural information, we fully excited the 1st transfer lens of the image aberration corrector in our microscope, which was never used in Ltz-STEM mode. As depicted in Fig. 1c, the lens is positioned near (~3 cm to) the sample. This positioning allows the lens to capture highly scattered electrons before they are obstructed by the liner tube without the magnetic field acting on the sample. Lorentz lenses are commonly available in commercial TEMs that lack image aberration-correction systems but were not previously used in Ltz-STEM mode to reduce camera length. In uncorrected microscopes, the Lorentz lens occupies the same position and shares the design of the first transfer lens in the image corrector. This allows it to also serve the function needed for LA-Ltz-4D-STEM. As a result, the highest accessible diffraction angle in the setup is increased to about 25 mrad without introducing significant distortions (Fig. S1). Thereby, the attainable maximum structural information was enhanced from approximately about 9 Å (in the conventional field-free mode) to <0.8 Å (in the new mode).

Arrays of diffraction patterns containing both unscattered and highly scattered beams can be captured using 4D-STEM data acquisition with the lens setup. Figure 1e shows two typical diffraction patterns of amorphous materials (Fe-based metallic glass) and crystalline (SmCo) recorded in this mode, as well as exemplary data analysis. In the case of amorphous materials characterized by a set of diffuse rings due to the lack of long-range order (Fig. 1e, left), the diffraction rings provide information on the short/medium-range atomic arrangement, e.g., the inter-atomic distance and atomic coordination. For crystalline materials, the diffraction spots convey details of the crystal symmetry, lattice parameters, and orientation (Fig. 1e, right). Analysis of the whole diffraction pattern with both low- and high-angle signals enables simultaneous imaging of local structure and magnetic fields.

The in-plane component of the magnetic fields inside of the sample deflects the electron beam through the Lorentz force as schematically depicted in Fig. 1d (Fig. S2a–c for the realistic case). Therefore, for simultaneous strain and magnetic field imaging, the center position of the diffraction pattern at each probe position is used to measure the direction and strength of the local in-plane magnetic field that the electron probe has passed through. In parallel, stress-induced distortions of the atomic structure of amorphous materials reflected as the elastic strain field are determined from the azimuthal elliptic distortion of the diffraction rings (Fig. S2d–g) in the 4D-STEM data and comparing it with a reference area, employing the data analysis process established by Gammer et al.[35] and Kang et al.[22]. The diffraction intensity in the radial direction of the diffraction pattern reflects the distances between atoms in the nanoscale volume, giving rise to information on the atomic density under the approximation that no considerable chemical variations are present. Following previous works[36,37], we estimated the relative atomic density by calculating the area encircled by the 1st ring of each diffraction pattern in the 4D-STEM data.

We used a deformed $Fe_{85.2}Si_{0.5}B_{9.5}P_4Cu_{0.8}$ (at.%) amorphous metallic alloy (metallic glass) with saturation magnetostriction[38,39] of $l_s \sim +40 \times 10^{-6}$ as an example in this study[10,21,40]. The alloy was deformed by scratch testing under ambient conditions using a diamond tip. As a result, shear offsets occurred on the scratched surface. Figure 2a

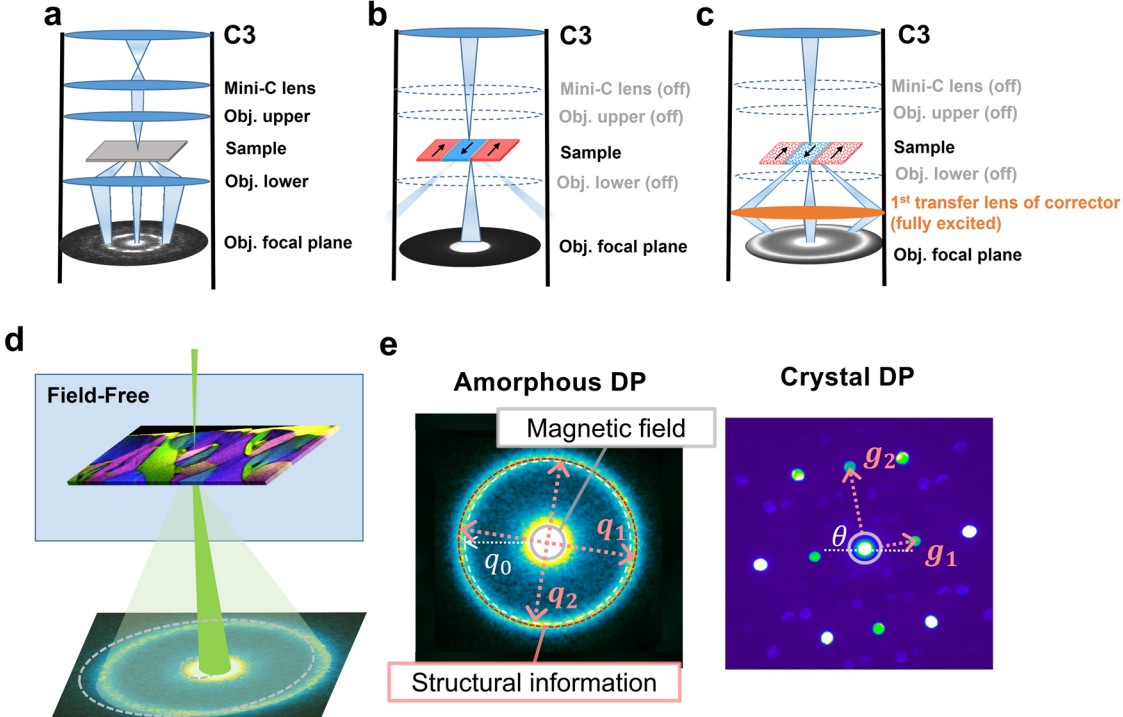

**Fig. 1 | Schematic illustration of LA-Ltz-4D-STEM and its optical setup. a–c** The optical setup using in nanobeam electron diffraction in standard mode with the objective lens on, conventional field-free Ltz-STEM, and LA-Ltz-4D-STEM. **d** The electron probe is focused on the TEM sample in LA-Ltz-4D-STEM. Spatially-resolved diffraction patterns are collected during scanning over an area of interest using 4D-STEM. **e** Exemplary data taken in the mode: left shows an amorphous diffraction pattern (DP) acquired from a Fe-based amorphous alloy, and right shows a crystalline DP acquired from a nano volume of a SmCo magnet. The position of the direct beam measures the momentum transfer by the Lorentz force, reflecting the local magnetic field of the sample. Structural information can be obtained from the diffraction ring (with details in Supplementary note 1) by calculating the relative atomic density from the area of the first ring, principal strains ($\vec{\varepsilon}_1$ and $\vec{\varepsilon}_2$) from its elliptical distortion ($\vec{q}_1$ and $\vec{q}_2$), and the structure factor S(q) through azimuthal integration and background subtraction. A PDF can be derived from the Fourier sine transformation of S(q) at each scan position. Moreover, the LA-Ltz-4D-STEM of crystalline materials can provide information on crystal symmetry, orientation, lattice parameters, etc.

shows an ADF-STEM image of the TEM lamella, where the location of shear bands can be estimated based on the shear offset at the surface.

We acquired a LA-Ltz-4D-STEM map in the area (green rectangle) including the two shear bands indicated by the red arrows, with a step size of 10 nm for balancing the field of view and data size. Figure 2b–d shows the LA-Ltz-4D-STEM results: maps of the magnetic field ($\vec{B}$), compressive strain ($\vec{\varepsilon}_{com}$), and relative density ($\Delta\rho$). We further determined the local pair distribution function (PDF) at each scanning position providing information on the atomic configuration using the same dataset as shown in Fig. S4 and Supplementary statement 3. In this work, we focused on the relative atomic density to describe the atomic packing for ease of presentation. The brightness in the $\vec{B}$ and the $\vec{\varepsilon}_{com}$ images represent the strength of the fields, and the colors represent their orientation. Intricate magnetic nanostructures are observed in the vicinity of the shear bands. They are clearly different from the magnetic domain structure of the undeformed sample (Fig. S5), which exhibits larger homogeneous domains (>3μm × 3μm).

Figure 2c visualizes that the strain concentrates near the shear bands with an orientation difference of ~ 90° at each side of the shear bands. The asymmetrical strain fields with a sharp transition across shear bands are in line with the previous strain observations in deformed amorphous metallic alloys[22,41,42]. The strain also induces the variation of the relative atomic density $\Delta\rho$ which reflects the net volume change due to the hydrostatic stress. $\Delta\rho$ suffers a sudden change from positive to negative across the shear plane (Fig. 2d), namely the pop-in side (where the surface of the material was pressed down) suffers a mainly compressive strain, while on the opposite, the pop-out side (the material surface was pushed out) suffers tractive

force that induces material dilatation. The strain field gradually fades out away from the shear bands in both compressed and tensile regions.

To rule out that density variations can introduce a phase shift of the exit electron wave (hence inducing a beam tilt) resulting in artifacts in the magnetic image, we conducted a conventional 4D-STEM measurement at the same sample position with the objective lens on (in a fully out-of-plane magnetized state due to the strong magnetic field generated from the objective lens). None of the contrast variations across the shear bands was visible in the DPC image (Fig. S6). The local density gradients are too small to contribute any disturbance to the magnetic image and the observed features in Fig. 2b reflect the pure magnetic field of the sample.

For materials with isotropic magnetostriction properties, the magnetoelastic energy density can be written as $e_{ME} = -\frac{3}{2}\lambda_s \sum_{i=1}^{3} \sigma_i \gamma_i^2$ [40], where $\lambda_s$ is saturation magnetostriction, $\sigma_i$ is the deviatoric strain and its in-plane component can be quantified through the strain measurement (Fig. S7a), $\gamma_i$ is the sine of the misorientation angle between $\vec{B}$ and $\vec{\varepsilon}_{ten}$. Taking advantage of the correlative imaging, a map of $e_{ME}$ can be obtained if the strain and magnetic vectors were measured in 3D by involving the tomography strategy for vector field[17,43] in our approach. As a simplified illustration, Figure S7b shows the map of the in-plane contribution of $e_{ME}$. As the positive saturation magnetostriction (~40 ppm), the $e_{ME}$ tends to align magnetic moments parallel to the tensile strain direction and perpendicular to the compressive strain direction.[12, 154] For magnetically soft ferromagnetic materials, the physical size of the domains is usually large to minimize the density of domain walls[7]. However, a high (magnetostatic) energy configuration is observed around the shear bands: small and periodically arranged magnetic domains are present

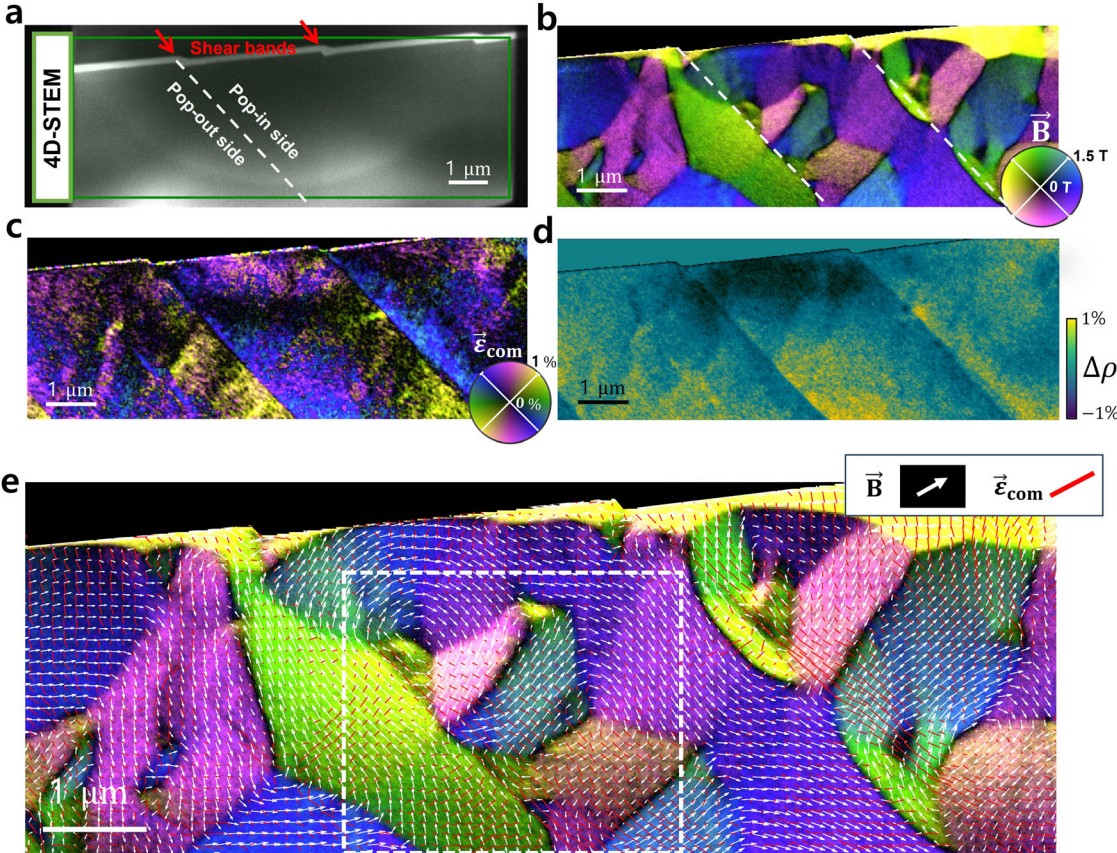

**Fig. 2 | Visualization of the magnetic field and atomic strain coupling using LA-Ltz-4D-STEM observation of a plastically deformed amorphous metallic alloy.** **a** STEM-ADF image of a TEM lamella, where the shear offsets are indicated by red arrows at the sample surface. The curved feature observed in the ADF-STEM background results from thickness variations due to the FIB-TEM lamella preparation (see Figure S3 and Supplementary note 2), and is unrelated to the shear band features. A LA-Ltz-4D-STEM map was acquired at the area indicated by the green rectangle. **b** Magnetic field ($\vec{B}$) image created using the principle of DPC imaging, but based on measuring the shift of the center of the 1$^{st}$ diffraction ring.

The color corresponds to the orientation, and the brightness corresponds to the amplitude of the fields as indicated by the color wheel. The locations of two shear bands are indicated by red dashed lines. **c** Strain map visualizing the compressive strain ($\vec{\varepsilon}_{com}$). The strain orientation is presented by a two-fold symmetrical color wheel, the brightness corresponds to the strain amplitude. **d** relative density ($\Delta\rho$). Yellow color represents high density and dark blue color low density.
**e** Simultaneous visualization. The local $\vec{B}$ field is represented by white arrows and the local $\vec{\varepsilon}_{com}$ field is represented with red sticks. Both field representations are overlayed on the image shown in Fig. 2b.

on the pop-in side of the shear band. They are confined by the strain-induced magnetic anisotropy perpendicularly aligned to the shear band orientation due to strong compressive strain parallel to the shear band (strong tensile strain perpendicular to the shear band due to Poisson's effect). In contrast, on the pop-out side of the shear bands, the magnetic structure is simpler and aligned parallel to the shear band direction due to the tensile strain parallel to the shear bands. Similar magnetic domain structures have been discussed in samples with orthogonal magnetic anisotropies in soft magnetic amorphous thin films induced by ion irradiation[14] and locally induced stress variation[13].

LA-Ltz-4D-STEM enables pixel-to-pixel correlation of the magnetization and atomic strain. We plot both information together in Fig. 2e. The white arrows represent the local $\vec{B}$ vectors. The red sticks representing $\vec{\varepsilon}_{com}$ visualize the strain field ($\vec{\varepsilon}_{ten}$ and $\vec{\varepsilon}_{com}$ are perpendicular to each other). The arrows and sticks are overlaid on the magnetic field image. Figure 2e shows that the orientation of $\vec{B}$ and $\vec{\varepsilon}_{com}$ are well correlated (namely, white arrows are close to being perpendicular to the red sticks) in the strongly strained area, except at domain walls and vortices, showing that the magnetization is dominated by the local strain.

We statistically studied the magnetoelastic coupling in the highly strained regions around the shear band (Fig. 3a). Figure 3b and d show 2D histograms of $\vec{\varepsilon}_{ten}$ and $\vec{B}$, respectively, where the axes correspond to the horizontal and vertical components of $\vec{\varepsilon}_{ten}$ and $\vec{B}$ in the multi-

information map (Fig. 3a) and the color corresponds to the number of pixels counted for each map (Fig. S8 shows the histogram of $\vec{B}$ from the whole map). A two-fold symmetry of the strain and magnetic field due to the geometry of the shear band is observed. Figure 3c and e show the spatial distribution of the tensile strains and magnetic moments binarized according to their orientation (blue is for perpendicular and light gray for parallel to the shear band). The major magnetic domains are coherent with the strain field. However, interestingly, a nontrivial part of the magnetic moments breaks the common expectation of the magnetoelastic coupling and deviated from the strain-induced anisotropy.

We statistically analyzed the deviation of the two types of magnetic moments in Fig. 3f. It is a 2D distribution plot (2D histogram) of $|\vec{B}|$ and the misorientation angle between $\vec{\varepsilon}_{ten}$ and $\vec{B}$. The high population of the pixels concentrates at small misorientation angles (<30°), forming a cluster ①, where the in-plane component of $\vec{B}$ increases linearly with decreasing misorientation between $\vec{\varepsilon}_{ten}$ and $\vec{B}$. This indicates that magnetic moments in cluster ① are governed by the in-plane strain-induced anisotropy. It tilts the moments in-plane, resulting in a stronger strength of the projected magnetic field. Figure 3g visualizes the spatial distribution of these magnetic moments. Another set of magnetic moments is aligned vertically in Fig. 3f and grouped as clusters ②. These moments are independent of the strain orientation, revealing the break of the magnetoelastic coupling. Figure 3h

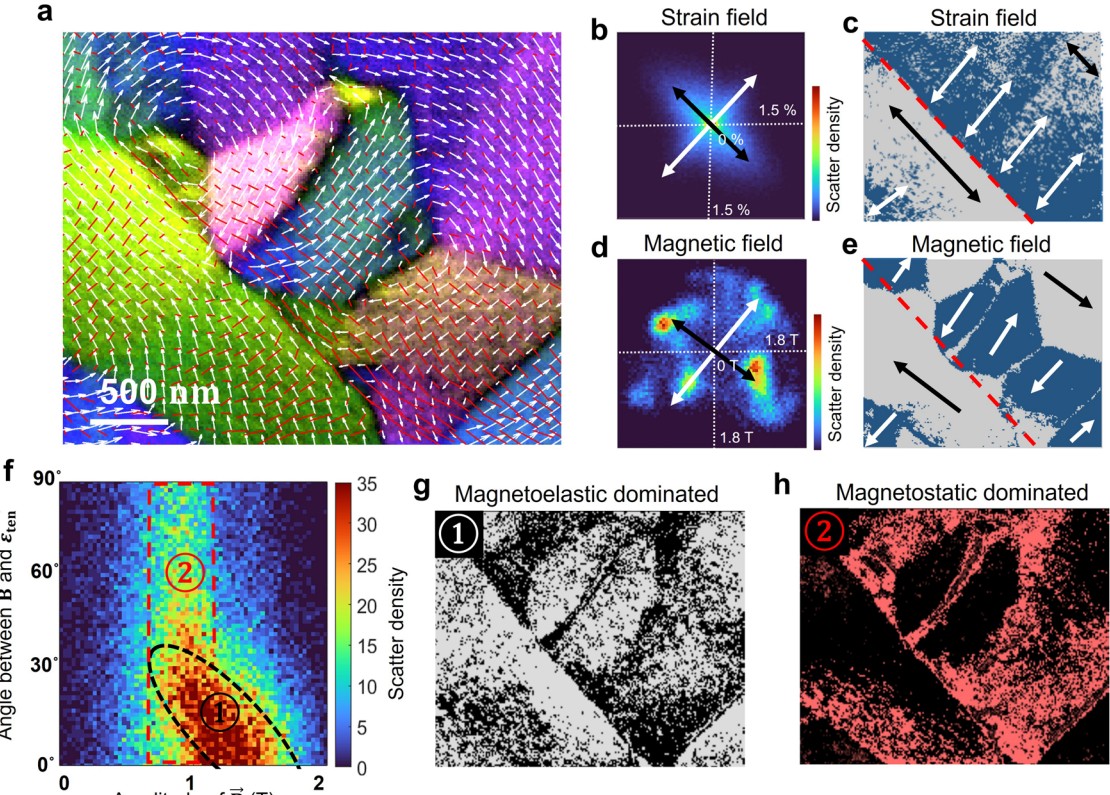

**Fig. 3 | Statistical analysis of the magnetic and strain coupling in the deformed amorphous metallic alloy using the LA-Ltz-4D-STEM data. a** Co-visualization of magnetic and strain field: a magnified image of the area indicated by the white dash rectangle in Fig. 2e. **b** 2D histogram of $\vec{\varepsilon}_{ten}$. The x and y axes correspond to the horizontal and vertical components of the tensile strain measured at each pixel in the map and the color corresponds to the number of pixels. The color corresponds to the number of pixels. Black and white arrows highlight the principal orientations. **c** Binarized maps visualizing the spatial distribution of $\vec{\varepsilon}_{ten}$ following the two principal directions in (**b**). Black and white arrows highlight the local major

orientations. **d** 2D histogram of $\vec{B}$ following the same approach as in **b**. Black and white arrows highlight the principal orientations. **e** Binarized maps visualize the spatial distribution of $\vec{B}$ following the two principal directions in (**d**). Black and white arrows highlight the local major orientations. **f–h** Correlative analysis of local strain and magnetization. **f** is a 2D histogram of two physical quantities, in which the horizontal and vertical axes are $|\vec{B}|$ and the misorientation angle between $\vec{\varepsilon}_{ten}$ and $\vec{B}$. (**g**, **h**) are the spatial distribution of the map contained in the components ① (white pixels) and ② (red pixels).

visualizes the spatial distribution of these moments, which contribute to the closure domains, domain walls and vortices with more complex magnetic structures. The existence of cluster ② is attributed to energetic competition between the strain-induced anisotropy (magnetoelastic) and the dipole-dipole interaction (magnetostatics). The closure domains extend over both regions to reduce the magnetostatic contributions of the magnetic domain state.

Interestingly, the amplitude of the in-plane magnetic components exhibits significant variations in different domains (Fig. S6a). For example, the in-plane $|\vec{B}|$ in Region ② is weaker than that in Region ①. The exact reasoning is not clear from our images, but this observation implies additional out-of-plane magnetization contributions to minimize the magnetization energy within the small bar-shaped sample. Especially, the complex patterns observed in the closure domain regions suggest a possible supplementary magnetic structure that varies in the sample thickness direction due to additional flux closure between the submicron domains. This could contribute to the observed decrease in the net in-plane magnetic flux density in some domains.

As shown above, LA-Ltz-4D-STEM maps enable the analysis of multiple physical quantities. In the current study, these are the in-plane $\vec{B}$ field with two degrees of freedom and in-plane strain tensor with 3 degrees of freedom (symmetric $2 \times 2$ matrix, in the above results represented in the form of two perpendicular principal strains) which can also be transformed to different aspects such as deviatoric and volumetric strain. Owing to the pixel-to-pixel correlation between the

different properties, there are many possibilities to correlatively analyze these quantities to uncover hidden physical phenomena. In addition to Fig. 3f–h, more examples are demonstrated in Figs. S9 and S10. A direct correlative investigation of domain wall properties is illustrated in Fig. S11.

Figure 4 shows an in-situ magnetization test for the same TEM sample (Supplementary Movie 1). The lamella was tilted to 10 degrees to apply a true in-plane magnetic field to the sample. The objective lens is gradually excited from 0 % to 9%, corresponding strength of the magnetic field in the sample plane is from 0–30 mT (estimated by the fact that the 100% objective lens produces a 2 T magnetic field). While the major magnetization in the sample is oriented along the external in-plane magnetic field ($\vec{B}_{in}$), the areas exhibiting complex domain patterns are strongly resistant to change, especially when compared to the undeformed sample (Supplementary Movie 2). We note that the magnetic domain structure almost recovered back to the original magnetization state after the objective lens current was set back to zero at the end of the test (Fig. 4g and h), except for only slight variations in some domain walls and vortices. This suggests minimal magnetic hysteresis and high reproducibility of the magnetization state in the material. The magnetic domain patterns are determined by strain-induced magnetoelastic anisotropy, offering potential applications for nanomagnetic devices.

The LA-Ltz-4D-STEM technique maps diffraction patterns with sub-angstrom information in reciprocal space in a field-free

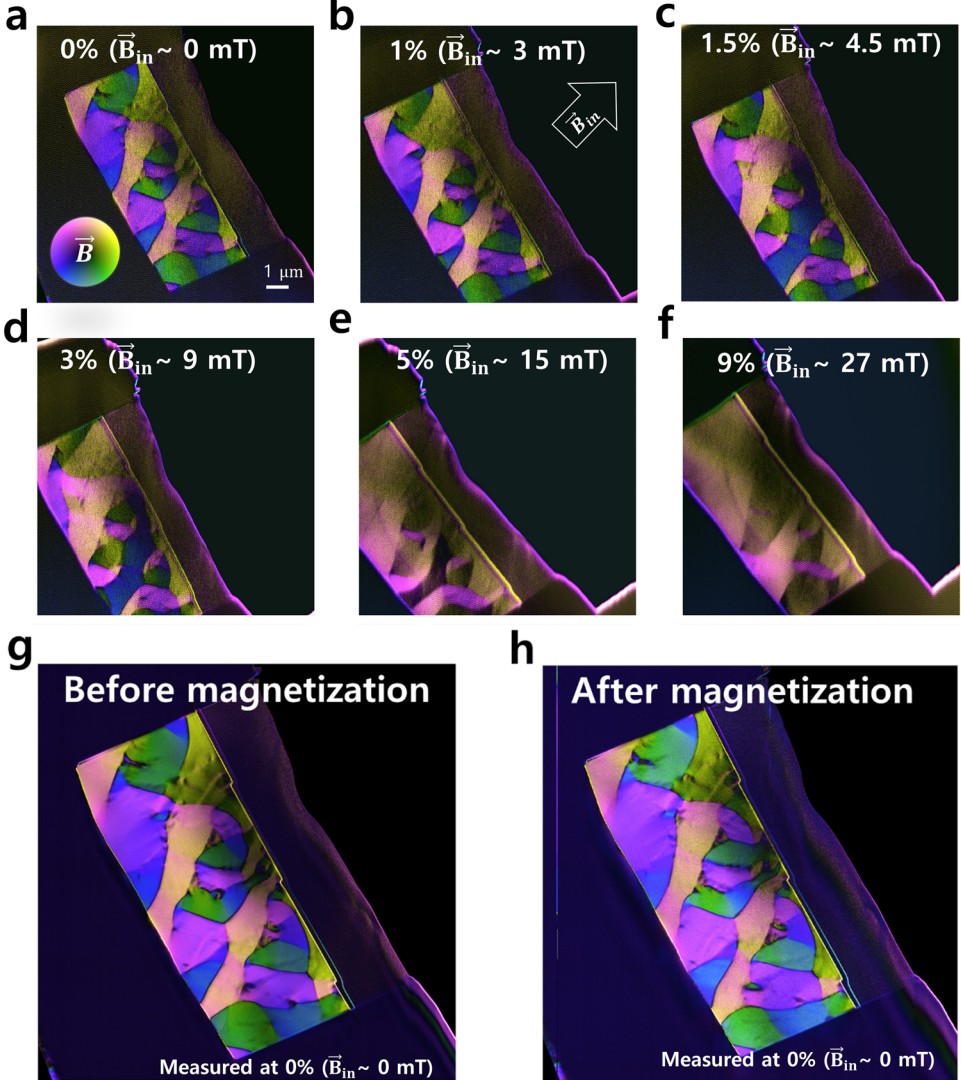

**Fig. 4 | In-situ magnetization test of a plastically deformed amorphous metallic alloy by activating the objective lens.** The magnetic field is imaged using a 4-segment detector in a conventional DPC setup. The sample is tilted to 10 degrees to apply an in-plane magnetic field to the TEM lamella. The objective lens was exited from (**a**) 0 % to (**f**) 9 %, corresponding to the strength of the magnetic field in the sample plane -0–30 mT. The inset arrows in (**b**) indicate the in-plane direction of the external magnetic field $\vec{B}_{in}$. **g**, **h** are high-quality maps before and after In-situ magnetization test recorded with longer exposure time.

condition and across an extensive field of view that can be easily expanded, mainly limited by hardware data storage. These enable simultaneous imaging of magnetic fields, atomic strain, packing density, and pair distribution function at the nanoscale. The result provides spatial information on multiple physical properties fully correlated at the pixel level, which enables comprehensive visualization and statistical evaluation of strain-induced effects on nanoscale magnetism. Using the method, we studied the strain-induced magnetic behavior in a shear-banded region of a deformed ferromagnetic amorphous alloy. The correlative image of the anomalous magnetic structure and the complex strain state, which is distributed on the nanoscale far beyond the uniaxial condition, explained the origin of the observation and clarified the magnetoelastic and magnetostatic competition. This methodology can also be applied to crystalline magnetic materials providing crystal symmetry, orientation, and phase analyses along with the magnetic information. Overall, LA-Ltz-4D-STEM breaks the information constraint of electron microscopy in studying nanostructured magnetic materials and enables direct visual correlation of structure and magnetic properties on the nanoscale.

## Methods

### Sample preparation

$Fe_{85.2}Si_{0.5}B_{9.5}P_4Cu_{0.8}$ (at.%) master alloy ingots were prepared from the melt by rapid solidification on rotating Cu wheels at Vacuumschmelze GmbH & Co. KG resulting in a ribbon width of about 25 mm and a thickness of about 20 μm. Scratch tests were performed with a scratch length $l_s = 1$ mm with normal loads $F_n = 10$ N for the Fe-based metallic glass ribbon with a sliding velocity $v_s = 0.1$ mm/s at ambient conditions using a diamond tip with a radius of 210 μm. For the STEM investigation, TEM lamella was prepared by FIB (FEI Strata 400S) from the ribbon at the left side of the scratch and 0.3 mm away from the scratch end. Thinning was performed to a sample thickness of about 200 nm for electron transparency at an acceleration voltage of 30 kV with gradually decreasing beam currents from 8 nA to 2 pA to reduce the ion beam damage.

### LA-Ltz-4D-STEM experiment

The development of the large-angle Lorentz 4D-STEM (LA-Ltz-4D-STEM) mode has been conducted using a Themis Z double-corrected TEM (ThermoFisher Scientific) operated at 300 kV. We first set the

microscope to STEM mode and the objective lens to zero current. The residual magnetic field was found to be between 3–5 mT measured using a Hall effect holder. This translates to an in-plane magnetic field of 0.5–0.86 mT when the sample is tilted by 10 degrees. The residual field becomes undetectable after one demagnetization cycle during that measurement, and we applied this process in the current work. Then, the 1st transfer lens of the image aberration-corrector in the microscope was fully excited. The two hexapole lenses of the corrector were switched off to eliminate any 3 and 6-fold distortion. For microscopes without an image-aberration corrector, the configuration is even simpler. Instead of the 1st transfer lens in the corrector, the Lorentz lens can be used to realize the LA-Ltz-4D-STEM mode. The diffraction and projection lenses are adjusted to image the post-sample diffraction plane to a camera located in the projection chamber with minimal distortions and optimized magnification of the diffraction pattern (i.e. the camera length) to fit the size of the camera. The approach can be used for all kinds of commercial TEMs equipped with either a Lorentz lens or image-aberration corrector and operated at all typical high tensions. After setting up the microscope mode, an electron probe with a small semi-convergence angle (in our case it is 0.26 mrad and limited by the uncorrected spherical aberration of the condenser system) is focused (to ~5 nm diameter in our case) on an electron transparent sample. The probe size is limited by the spherical aberration of the probe-forming lens. For the probe-aberration corrected Ltz-STEM, it can be pushed to a half nanometer. The field-free condition was double-checked by observing the domain structure of a TEM lamella lift-out from the undeformed area of the Fe-based amorphous metallic alloy. For 4D-STEM data acquisition, we used a OneView camera (Gatan Inc.). The magnification of the diffraction pattern (i.e. camera length) was set to capture the first diffraction ring with a large diameter spanning >80% range of the camera to enhance the sensitivity and accuracy for measuring the distortion. 4D-STEM maps were acquired by scanning the electron probe over a 2D sample plane with a step size of 15.8 nm and a frame size of $620 \times 225$ and camera binning to $256 \times 256$ pixel$^2$ for balancing the storage space in our camera computer with an exposure time of 3.3 ms per frame (frame rate of ~300 f/s). The central position shift in the diffraction pattern during scanning was resolved through precise alignment of the scan pivot point and the Descan function. This improvement was achieved by eliminating the significant optical distortion inherently caused by the objective lens. A remaining diffraction shift can be corrected by post-background processing by capturing a reference map without the sample.

### Data processing

The magnetic fields inside the sample deflect the electron beam due to Lorentz force. The deflection of the beam is a measure of the direction and strength of the local magnetic field, which contains the 1st order gradient of the phase of the electron wave exiting from the specimen, creating so-called differential phase contrast (DPC). The in-plane magnetic field can be calculated as $\vec{\mathbf{B}} = \frac{h}{e\lambda t}\theta_{deflect}$, where $\theta_{deflect}$ is the beam deflection angle, $h$ Planck's constant, $\lambda$ the relativistic electron wavelength, $e$ the charge of an electron, and $t$ the sample thickness, which can be estimated from a thickness map (Fig. S3) obtained by electron energy loss spectroscopy (EELS). The diffraction of a typical amorphous material shows a diffuse ring pattern (Fig. 1e). The center of the diffraction ring can be used to measure the beam deflection providing higher accuracy than using the transmitted center beam directly. While the shortened camera length in principle reduces the sensitivity for measuring the primary beam deflection, using the shift of the first diffraction ring to create the DPC signal successfully provided good quality in imaging the magnetic field in the sample as shown in the following section. We measured the standard deviation of magnetic variation within a magnetic domain of the ferromagnetic amorphous alloy used in this study, recording values as low as 10 mT

for the direct beam and 70 mT for the first ring. Due to the local structure and probably also the magnetic variation in the sample, this value is an overestimate and upper boundary of the real measurement accuracy. Nevertheless, these values are considerably smaller than the overall magnetization (~1 T) within the domain. To prevent damage to the camera from the direct beam, a beam stopper was used, and the first ring was utilized for the experiment.

The local stress in the metallic glass induces a structural anisotropy, which results in an elliptic distortion of the diffraction ring leading to a deviation from the ideal circle as illustrated in Fig. 1e (The diffraction pattern was further elongated artificially for easy presentation). Therefore, the strain can be mapped by determining the ellipticity of the diffraction ring in each local diffraction pattern of the LA-Ltz-4D-STEM dataset (with details in Supplementary note 1). In this work, we use the convention that the first principal strain is oriented along the compression direction providing the compressive component ($\vec{\varepsilon}_{com}$), which in turn results in the tensile component ($\vec{\varepsilon}_{ten}$) as the second principal strain that is oriented perpendicular to the first principal strain. We also quantify the deviatoric strain from the strain components to disentangle the true local distortion of the material and the local net volume change as a response to the local hydrostatic stress. We estimated the relative atomic density by calculating the area encircled by the 1st ring of each diffraction pattern in the 4D-STEM data. This approach takes the elliptical deviation of the diffraction rings resulting from deviatoric strain into account and is an intuitive way to analyze the volumetric strain and disentangle the density information from the sample thickness variation.

The method can also be applied to crystalline magnetic materials. Figure 1e (right) shows an example diffraction pattern recorded from a SmCo magnet in the large-angle field-free mode, where crystal symmetry, orientation, and phase analyses can be performed using the 4D dataset. Of course, dynamic electron scattering issues should be considered for the thick crystal samples, but applying beam precession has been shown to significantly reduce the crystalline artifacts suffered in the conventional DPC[44,45]. Padgett et al. recently demonstrated that the exit-wave power-cepstrum method, a reverse Fourier transform of the logarithm of diffraction, can successfully eliminate the dynamic scattering induced artifact in a strain map[46].

### Data availability

The data generated in this study are available in the KITOpen database under accession code https://doi.org/10.35097/ms36bzm0nrrzj51g. The data are also available upon request from the authors.

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

## Acknowledgements

The authors express their gratitude to the Karlsruhe Nano Micro Facility (KNMFi) for their assistance and for providing access to FIB and TEM facilities. X. M. acknowledges financial support from the Deutsche Forschungsgemeinschaft (DFG) funding (MU 4276/1-1) and National Natural Science Foundation of China (127000- 832011). S.K. and C.K. acknowledge the support from collaborative research centre FLAIR (Fermi level engineering applied to oxide electroceramics), which is funded by the German Research Foundation (DFG), project-ID 463184206—SFB 1548. This work was supported by the Technology Innovation Program (RS-2024-00418991) funded By the Ministry of Trade, Industry & Energy (MOTIE, Korea). Additionally, the authors appreciate the support received from the Joint Lab Model Driven Materials Characterization (MDMC) and acknowledge the backing from the Helmholtz Imaging Project (HIP) BRLEMM. R.E.D.-B. is grateful for funding from the Deutsche Forschungsgemeinschaft (DFG, German Research Foundation) – Project-ID 405553726–TRR 270.

## Author contributions

S.J.K. and X.M. developed the methods and performed the TEM experiments. K.D. and C.M. provided the Fe-based sample. S.J.K. and

A.C. performed the deformation experiments. S.J.K., M.T., X.M., D.W., R.D.-B., J.M., and C.K. analyzed and discussed the data. S.J.K., X.M., and C.K. wrote the manuscript. All authors contributed to the revision of the manuscript.

## Funding

## Competing interests
The authors declare no competing interests.

## Additional information

[1]Institute of Nanotechnology (INT), Karlsruhe Institute of Technology (KIT), 76344 Eggenstein-Leopoldshafen, Germany. [2]In-situ Electron Microscopy, Department of Materials Science, Technical University of Darmstadt (TUDa), 64287 Darmstadt, Germany. [3]Karlsruhe Nano Micro Facility (KNMFi), Karlsruhe Institute of Technology (KIT), 76344 Eggenstein-Leopoldshafen, Germany. [4]Physical Metallurgy, Department of Materials Science, Technical University of Darmstadt (TUDa), 64287 Darmstadt, Germany. [5]Laboratory for High Performance Ceramics, Swiss Federal Laboratories for Materials Science and Technology (Empa), 8600, Dübendorf, Switzerland. [6]Korea University of Technology and Education (Koreatech), 330708 Cheonan, Republic of Korea. [7]Herbert Gleiter International Institute, Liaoning Academy of Materials, Shenyang 110167, China. [8]Ernst Ruska-Centre for Microscopy and Spectroscopy with Electrons, Forschungszentrum Jülich GmbH, 52425 Jülich, Germany. [9]Nanoscale Magnetic Materials – Magnetic Domains, Department of Materials Science, Kiel University, 24143 Kiel, Germany. [10]Kiel Nano, Surface and Interface Science (KiNSIS), Kiel University, 24118 Kiel, Germany. [11]School of Materials and Energy and Electron Microscopy Centre, Lanzhou University, Lanzhou 730000, China. ✉e-mail: christian.kuebel@kit.edu; muxk@lzu.edu.cn

