## [Transparent Peer Review file · Nature Communications]

Large-angle Lorentz 4-Dimensional Scanning Transmission Electron Microscopy for Simultaneous Local Magnetization, Strain and Structure Mapping

Corresponding Author: Dr Sangjun Kang

A version of this paper was originally rejected for publication by Nature Communications, however that decision was reconsidered after appeal by the authors.

Version 0:

Reviewer comments:

Reviewer #1

(Remarks to the Author)

This paper reports a new method in 4D-STEM that enables simultaneous mapping of magnetic and atomic structures. The most critical key point is to use a Lorentz lens below an objective lens to extend the upper limit of diffraction angles and acquire a large-angle diffraction pattern showing the sample's atomic arrangement structure with a 4D-STEM detector. However, this paper cannot impact the readers of "Nature Communications" because the idea is not innovative, and the Lorentz lens is not the author's original (it can be supplied by a TEM manufacturer). Some beautiful experimental results are shown and clearly explained, but they are just examples demonstrating the method's capabilities. Therefore, this paper is recommended for publication in other journals.

(Remarks on code availability)

Reviewer #2

(Remarks to the Author)

This manuscript develops a method for the simultaneous measurement of local strain and magnetization, demonstrating it using a magnetic metallic glass sample. The method is novel and could become a powerful analytical tool for magnetic materials. I have some concerns, but if these are addressed, I would recommend the paper for publication in Nature Communications.

The authors discuss the optics of TEM in Lorentz mode in detail. However, I do not find this discussion to be particularly novel, nor likely to interest many readers. This technical detail should be moved to the Methods section.

The authors extract both the magnetic field signal and the strain signal from the diffuse ring. For the magnetic field signal extraction, they assume a rigid shift, which is generally incorrect because the intensity distribution changes in the bright field disk. Similarly, the magnetic field does not rigidly shift the diffuse ring but rather alters the intensity distribution within the ring, potentially affecting the strain analysis. Therefore, the observed correlation between magnetization and strain might be an artifact. The authors need to carefully discuss the effect of the magnetic field on the diffuse ring to demonstrate the validity of their method. Additionally, to justify their demonstration results, they should perform strain measurements under varying magnetization conditions to show that the correlation is not an artifact.

The authors assume that the magnetic field is 0 mT when the objective lens is off. However, unless a demagnetization process is conducted, there could be residual magnetic fields present. I believe the objective lens power supply is monopolar, so AC demagnetization is not possible. While I do not expect this to significantly affect the experimental results,

the text should be described more accurately. Ideally, the residual magnetic field should be measured directly. If the authors do not have a method to measure this directly, they could estimate the magnitude by tilting the non-deformed sample with the objective lens off and observing changes in the magnetic domain structure.

(Remarks on code availability)

Reviewer #3

(Remarks to the Author)

The manuscript by Kang et al. presents a study on magnetoelastic coupling in a Fe-based amorphous alloy soft magnet using a new method, large-angle Lorentz 4-dimensional scanning transmission electron microscopy (LA-Ltz-4D-STEM), enabling simultaneous acquisition of magnetization and strain maps at nanometer scale. Key findings of this study are:

- Development of LA-Ltz-4D-STEM by employing Lorentz lens in the conventional magnetic field-free TEM, 3 cm distance away from the sample to shorten the camera length, thus enlarging the collection angles in the detector plane.
- LA-Ltz-4D-STEM demonstrates direct observations of a high energy magnetostatic configuration, nanoscale dense magnetic domains, around the shear bands in a soft ferromagnetic matter.
- By conducting statistical analysis and applying external magnetic fields, strain-magnetism coupling was confirmed.

Unlike most electron microscopy studies of magnetic materials, this study presents a straightforward yet exemplary methodology for correlating local strain with magnetic domain structures. It also demonstrates well-designed and controlled sample preparation and data processing/analysis for magnetization and strain mapping. The manuscript is well written and experimental data are clearly presented to demonstrate the magnetoelastic coupling. However, it is unclear whether the authors developed and installed new hardware for LA-Ltz-4D-STEM or simply found a customized lens configuration to shorten the camera length using existing lenses. Since the sample studied was amorphous and the data analysis was relatively insensitive to diffraction conditions, it is uncertain if this technique can be applied to other ferroics in crystalline phases. Additionally, while the authors claim that atomic structures correlating with magnetic structures can be studied, they only provide strain information. Furthermore, the phenomenon of strain-induced magnetic anisotropy is relatively well known (is this new for this magnetic material?) and not particularly surprising. I think that the authors should justify their findings as a significant advancement to warrant publication in Nature Communications.

Detailed comments

1. It is unclear whether the authors developed and installed new hardware for LA-Ltz-4D-STEM or simply found a customized lens configuration to shorten the camera length using existing lenses. If the former, I would like to recommend that the authors elaborate more in detail about the new lens configuration and installation. In either case, are there any changes in magnetic fields on the sample plane?
2. It is interesting to measure the phase shift of the electron beam using the center of the 1st amorphous Bragg ring. Quantifying the center-of-mass (CoM) shift of a transmitted and diffracted beam from crystalline samples using a nanoscale electron probe illuminating multiple atomic columns is often complicated due to the dynamic scattering process. If the authors provide full details on CBED pattern data processing such as CoM shift, 1st Bragg ring detection, measuring ellipticity, etc., it will be really useful for people who are interested in this method.
3. In the Figure 2a (STEM ADF image) shows a curved shape feature in the sample. Similar curved features are also can be seen in the EFTEM image shown in Figure S3. It seems that the relative density map (Figure 2d) shows that reduced density can be seen near the sample edge and between two pop-in/outs. The strain changes across the pop-in and pop-out shear bands also appear to be suppressed in the reduced density area. Can authors more elaborate on this?
4. In Figures 2, 3, and 4, most important data showing the magnetoelastic coupling, there are clear differences in magnetic domain structures between the pop-in and pop-out regions. In the pop-out region, compressive strain perpendicular to the shear band direction and tensile strain parallel to the shear band direction are observed. This trend seems to be easily understood by the conventional Poisson effect. In this respect, even though the pop-in region has a simple 90° rotation strain field relationship with respect to the pop-out one, why does the magnetic domain have much larger magnetic domains than those in the pop-out region?

(Remarks on code availability)

Reviewer #4

(Remarks to the Author)

The authors report a study of mapping magnetic spin textures and strain field of a Fe-based amorphous alloy simultaneously by using 4D-STEM in a Lorentz mode. By evaluating the center shift and distortion (ellipsoid shape) of the first diffraction ring, the magnetic induction, strain and atomic relative density maps can be retrieved. Along with the statistical analyses, the authors present the coupling between magnetoelastic and magnetostatic energies in this material. The field-driven behavior was also explored and the results show that the deformed alloy has low hysteretic behavior. This work is of interest and technically sound, and the paper is well-written. However, the measurement of magnetic induction based on DPC method has been well developed. The approach to estimate the strain field etc. have already been reported by other work Ref [29] and Authors' other publications Refs [23] [31]. The authors mention they have used the first transfer lens of the imaging Cs-corrector as the Lorentz lens for controlling the diffraction space information. This overall limits the applicability of the approach as the microscope needs a Cs-corrector and therefore cannot be applied to any commercial microscope. Using

mini-objective lens or Lorentz lens including use of Cs-corrector's first transfer lens also has been previously documented. Furthermore, the authors claim of correlating magnetic structure with atomic structure is a little overstated. At best, the current work can be considered correlating microstructure with magnetic domain structure. There isn't really atomic scale information presented or derived from the dataset. The insights derived into the correlation of magnetic domains and strain has been studied extensively using Fresnel mode LTEM. The authors work does not provide any novel insights into the coupling. Based on these remarks, the work/approach does not meet the required degree of novelty of significance in the field to be published in Nature Communications. Nevertheless, the Authors do provide a useful method to gain insight into the correlation between strain and magnetic domains, which would be potentially interesting for specific communities. Therefore, this work should be submitted to a more specialized journal.

There are several questions that the authors should consider before resubmission.

1. in the introduction, the statement, "The magnetic structure is intrinsically determined by the interatomic distances and coordination.", is not entirely correct. The magnetic structure is also strongly dependent on the chemical elements. Also, under the circumstances that are presented in the authors work, the more stronger contribution to the magnetic structure will be from shape anisotropy (magnetostatic energy) due to the thin TEM lamella.

2. What materials is the diffraction patterns taken from in the Figure 1b top? The Author could label clearly the materials for Figure 1b in the caption as I thought both diffraction patterns are from the $\text{Fe}_{85.2}\text{Si}_{0.5}\text{B}_{9.5}\text{P}_{4}\text{Cu}_{0.8}$. Figure 1b shows the diffraction patterns of crystal with multi-phases and list the pair distribution function, which however are not very relevant to current work based on the discussion in the paper and causes confusion. This information could be included in the supplementary or could just cite relevant paper. This is just a minor suggestion.

Why do the authors present the diffraction ring of SmCo not that of Fe-based alloy in Figure 1b bottom?

3. Were there additional configurational changes (access to lens currents, vendor specific hardware access) that were needed to achieve the higher diffraction angle in the setup presented?

4. Could author make red lines (representing compressive field) in Figure 2e more obvious, such as, increasing line thickness? It is hard to see them from a colorful background.

5. How does the relative density that the authors have calculated compare with the HAADF image intensity?

6. The authors mention they also performed imaging using conventional 4D-STEM imaging. If there is a strong magneto elastic coupling, introducing a relatively strong field and out of plane magnetization will affect the local strain and shear bands. How did the authors account for this?

7. The authors state that the magnetic moments are aligned parallel to the tensile strain direction. These results are not unexpected. Also such effects have already been demonstrated using Fresnel LTEM imaging where changes in domain wall and domain orientations can be easily correlated with diffraction patterns and therefore strain/crystallographic directions.

8. The authors state that the closure domains are formed to reduce magnetic anisotropy. This statement is incorrect, and the authors should be careful about this. Closure domains would be a results from the shape anisotropy rather than magnetic anisotropy.

9. Overall in the analysis of strain, there was no error bar stated. These measurements are based on slight changes in the distortion of the diffraction ring, and therefore can be very sensitive to the measurement, intensity, area on the detector. An associated error bar in the measurements should be included. Also what is the sensitivity level for the measurement of strain and magnetic induction map. With the higher diffraction angle, as the authors mention, the measurement of deflection is effectively smaller in the diffraction plane. Can the authors provide some comparison and how statistically significant the measurements are?

10. Considering that Figure 3 shows statistical analysis of the coupling between magnetic field and tensile field, it would be very helpful to understand the results by replacing the compressive field map (the red lines in Figure 3(a)) with tensile field map to avoid confusion.

11. The author should mark the external in-plane field value in Figure 4(a-f), which may allow to easily understand the figure. In addition, I would suggest to make it clear that the Figure 4g and h were recorded at zero field, as the Figure h seems showing the spin textures under an external field with the label of "After magnetization".

12. The nearly identical spin textures was achieved after removing magnetic field in the deformed sample. How about the undeformed sample? Have the authors seen the vortex restored when reducing the field strength to zero?

13. There is the lack of the discussion of strain-induced magnetic properties in the abstract, introduction or conclusion.

(Remarks on code availability)

(Remarks to the Author)

(Remarks on code availability)

Reviewer #6

(Remarks to the Author)

(Remarks on code availability)

Version 1:

Reviewer comments:

Reviewer #1

(Remarks to the Author)

Compared to the previous version, which tried to insist on the advanced method's novelty, the revised version focuses on the method's validity and potential by adding supporting data and discussing some concerns deeply according to the referees' comments). The meaning of the atomic structure also became clearer. As for my concern, the author explained that they developed the electron optics tuned to be adaptable for LA-Ltz-4D-STEM (not the hardware and software system itself), and the details were described in the "Method" section.

The current version is acceptable for publication.

Reviewer #2

(Remarks to the Author)

I have reviewed the revised manuscript and appreciate the authors' thorough and thoughtful responses to all of my concerns. The revisions have significantly improved the clarity and overall quality of the paper, making it more accessible to the readership.

At this stage, I have no further comments or suggestions. I believe the manuscript is suitable for publication in its current form.

Reviewer #3

(Remarks to the Author)

It seems the authors have made a thorough effort to address the reviewers' comments, and their responses are well prepared. I don't have further comments or questions about the experiments and interpretations, as I believe those points have already been sufficiently raised. However, I am not sure about the novelty of this work. Despite the beautiful experiments and extensive efforts demonstrated, I am not fully convinced by their justification of the work's novelty, also raised by other reviewers.

Reviewer #4

(Remarks to the Author)

The authors have modified the manuscript as per the suggested changes. They have highlighted the importance of their work on determining local variation of strain as well as magnetic domains. However, the title of their work is still misleading since simultaneous magnetic and structure imaging has been performed before. I would suggest updating the title along the lines of the importance of their work which is on local variations of strain/magnetization in the sample.

Reviewer #5

(Remarks to the Author)

Reviewer #6

(Remarks to the Author)

We sincerely thank the Reviewers for their recommendations and comments on our work. In this revision, we have addressed the Reviewer's comments, taken constructive suggestions, and revised our manuscript accordingly. The reviewers' comments are displayed in *italic format*. Our response is in black and changes in the main text and Supplementary Information are highlighted in red for easy further review.

To Reviewer #1:

This paper reports a new method in 4D-STEM that enables simultaneous mapping of magnetic and atomic structures. The most critical key point is to use a Lorenz lens below an objective lens to extend the upper limit of diffraction angles and acquire a large-angle diffraction pattern showing the sample's atomic arrangement structure with a 4D-STEM detector. However, this paper cannot impact the readers of "Nature Communications" because the idea is not innovative, and the Lorenz lens is not the author's original (it can be supplied by a TEM manufacturer). Some beautiful experimental results are shown and clearly explained, but they are just examples demonstrating the method's capabilities. Therefore, this paper is recommended for publication in other journals.

Reply:

Currently, neither electron microscope manufacturers nor existing studies offer a method for capturing structural information during magnetic imaging in commercially available S/TEMs. Most researchers studying the structure of magnetic materials must place their samples in a strong magnetic field of approximately 2 T, generated by the objective lens. It disrupts the original magnetic configurations and, therefore, makes a direct correlation between magnetic and materials nanostructures impossible, presenting a significant limitation in such studies.

Prior to our work, neither Lorentz TEM nor DPC could provide structural information due to their inherently limited spatial resolution under field-free conditions. Our study introduces a novel optical setup that, for the first time, allows to capture large-angle 4D-STEM under field-free conditions. This innovation enables simultaneous mapping of both magnetic fields and structural information. The ability to correlate these factors at the pixel level precision within a single experiment offers a highly sought-after tool for researchers and a novel approach to visual and statistical analysis of multidimensional physical quantities at the nanoscale.

Despite extensive study of the magnetoelastic phenomenon, the nanoscale effects of strain on magnetism remain poorly understood, as it has been challenging to simultaneously measure magnetization and atomic strain at this scale. Most experimental studies for strain-induced magnetic phenomena, such as those published in *Nature Communications* 2023, 14:3963 and *Nature Nanotechnology* 2015, 10(7):589-92, have focused on simple uniaxial strain conditions. However, strain often varies locally at the nanoscale, which is both common and relevant for nanomagnetism, but has eluded quantitative study until now.

The amorphous ferromagnetic alloy investigated here has drawn significant interest due to its combination of magnetic softness and pronounced magnetostriction, with applications like ultrasensitive

magnetic field sensors. The shear banding process, which is typical plastic deformation mechanism of the material, generates intricate strain fields around shear bands (*Advanced Materials* 2023, 35, 2212086) and alters the local magnetic configurations (*Nature Communications* 2018, 9:4414). However, these two physical phenomena had been studied independently, with no way to directly correlate them. As a result, the effect of local strain fields on magnetic configurations near shear bands has remained unclear, hindering a full understanding of how deformation impacts magnetic properties and limiting further advancements in material applications.

Our results correlatively visualize, for the first time, the correlation between nanoscale strain variations and abnormal magnetic configurations, as well as the competition between magnetoelastic and magnetostatic forces at the nanoscale. These findings and new method are expected to significantly contribute to ongoing research in magnetism, amorphous alloys, and advanced microscopy techniques, making them of great interest to readers of *Nature Communications*.

To Reviewer #2:

This manuscript develops a method for the simultaneous measurement of local strain and magnetization, demonstrating it using a magnetic metallic glass sample. The method is novel and could become a powerful analytical tool for magnetic materials. I have some concerns, but if these are addressed, I would recommend the paper for publication in Nature Communications.

Reply:

Thank you for your positive feedback on our manuscript and for recognizing the novelty and potential impact of our method. We appreciate your willingness to recommend our work for publication in *Nature Communications*. We address your concerns in detail below.

The authors discuss the optics of TEM in Lorentz mode in detail. However, I do not find this discussion to be particularly novel, nor likely to interest many readers. This technical detail should be moved to the Methods section.

Reply: The authors consider the electron optical setup as novel, but we agree that the details of this setup will only interest a limited number of readers. Therefore, we shortened the section on the “Development of simultaneous magnetic and atomic structure mapping” and moved most of it to the Method section.

The authors extract both the magnetic field signal and the strain signal from the diffuse ring. For the magnetic field signal extraction, they assume a rigid shift, which is generally incorrect because the intensity distribution changes in the bright field disk. Similarly, the magnetic field does not rigidly shift the diffuse ring but rather alters the intensity distribution within the ring,

potentially affecting the strain analysis. Therefore, the observed correlation between magnetization and strain might be an artifact. The authors need to carefully discuss the effect of the magnetic field on the diffuse ring to demonstrate the validity of their method. Additionally, to justify their demonstration results, they should perform strain measurements under varying magnetization conditions to show that the correlation is not an artifact.

Reply:

In atomic-level DPC experiments, intensity variations within the direct disk are often observed. This occurs when the probe size exceeds the range over which the variation of the atomic potential can be approximated as linear. As a result, the first-order gradient of the electron wave phase exiting the specimen is not constant within the probed region, leading to intensity redistribution.

However, in the case of ferromagnetic materials, the domains and their walls are much wider than the electron probe, causing a rigid shift of the direct disk rather than just intensity variations, as illustrated in Figure R1b. This rigid shift also applies to the diffraction ring of the amorphous diffraction pattern, as demonstrated in Figures R2a-c. Consequently, using the shift of the first diffraction ring to generate the DPC signal allows us to achieve high-quality imaging of the magnetic field in the sample. Additionally, Figures R2c and R2g illustrate that the magnetic and strain effects on the first diffraction rings belong to two different categories, operating independently of each other.

As per your suggestion, we compared measurements under different magnetization conditions, such as the fully saturated and unmagnetized states, as shown in Figure S5 of the original manuscript and displayed below as Figure R3 for convenience. Neither the magnetic domain features nor strain features are visible in the magnetic image in the fully saturated state where the strain still exists at the same level as the unmagnetized case, confirming that the sample strain does not introduce artifacts into the magnetic images. This demonstrates that using the shift of the first diffraction ring to measure the DPC signal effectively enables high-quality magnetic field imaging in the sample. The strain maps for both magnetization conditions are consistent, with slight variations potentially due to the magnetostriction effect. Importantly, the magnetic and strain signals remain independent and do not interfere with one another.

Figure R1. (a) schematic diagram showing the principle of magnetic imaging in STEM. (b) typical diffraction disks from two magnetic domains and their comparison by subtracting one from another, the black arrow points to the deflection of the electron beam caused by Lorentz force. (d) A typical magnetic induction (magnetization) map of FeRh film by using the DPC approach (registering the center of 4D-STEM data). The color and brightness represent the measured field direction and strength respectively, summarized by the inserted color wheel at the top right.

Figure R2: Diffraction patterns acquired by LA-Ltz-4D-STEM from different sample locations. **a** From a domain pointing left-up direction and **b** right-down direction. **c** Subtraction of both **a** and **b** diffraction patterns. **d** Unstrained case, and **f** strained case. **g** Subtraction of both **d** and **f** diffraction patterns

Figure R3: Map of the amplitude of the in-plane magnetization of the deformed amorphous $\text{Fe}_{85.2}\text{Si}_{0.5}\text{B}_{9.5}\text{P}_4\text{Cu}_{0.8}$ metallic alloy at **a** an unmagnetized state (field-free condition), and **b** vertically magnetically saturated state (conventional microprobe STEM condition with objective excited about 2 T). **c** Line profiles taken across the shear band. Line scan 1: From the image of **a** along the black dash arrow. Line scan 2: From **b** along the red dash arrow. Map of the amplitude of the tensile strain at **d** unmagnetized state (field-free condition), and **f** vertically magnetically saturated state. **g** Line profiles taken across the shear band. Line scan 3: From the image of **d** along the black dash arrow. Line scan 4: From **f** along the red dash arrow.

The authors assume that the magnetic field is 0 mT when the objective lens is off. However, unless a demagnetization process is conducted, there could be residual magnetic fields present. I believe the objective lens power supply is monopolar, so AC demagnetization is not possible. While I do not expect this to significantly affect the experimental results, the text should be described more accurately. Ideally, the residual magnetic field should be measured directly. If the authors do not have a method to measure this directly, they could estimate the magnitude by tilting the non-deformed sample with the objective lens off and observing changes in the magnetic domain structure.

Reply: We appreciate the reviewer's concern regarding the residual magnetic field and the demagnetization process. In our TEM (ThermoFisher Scientific Themis Z), the objective lens

can be operated between +100% and -10% magnetic field strength. By reversing the field, we were able to successfully demagnetize the lens.

The residual magnetic field without the demagnetization process was measured by a TFS engineer using a Hall effect holder during the installation of the TEM and was found to be between 3-5 mT. This translates to an in-plane magnetic field of 0.5-0.9 mT when the sample is tilted by 10 degrees. We also observed that the field became undetectable after one demagnetization cycle during that measurement, and we applied this process in the current work. In addition, similarly as suggested by the reviewer, the in-situ magnetization experiments in Figure 4 in the main manuscript ruled out any significant effect of the small residual magnetic field on our observation. Following your suggestion, we have added this information to the revised manuscript to describe it more accurately.

To Reviewer #3:

The manuscript by Kang et al. presents a study on magnetoelastic coupling in a Fe-based amorphous alloy soft magnet using a new method, large-angle Lorentz 4-dimensional scanning transmission electron microscopy (LA-Ltz-4D-STEM), enabling simultaneous acquisition of magnetization and strain maps at nanometer scale. Key findings of this study are:

- Development of LA-Ltz-4D-STEM by employing Lorentz lens in the conventional magnetic field-free TEM, 3 cm distance away from the sample to shorten the camera length, thus enlarging the collection angles in the detector plane.*
- LA-Ltz-4D-STEM demonstrates direct observations of a high energy magnetostatic configuration, nanoscale dense magnetic domains, around the shear bands in a soft ferromagnetic matter.*
- By conducting statistical analysis and applying external magnetic fields, strain-magnetism coupling was confirmed.*

Unlike most electron microscopy studies of magnetic materials, this study presents a straightforward yet exemplary methodology for correlating local strain with magnetic domain structures. It also demonstrates well-designed and controlled sample preparation and data processing/analysis for magnetization and strain mapping. The manuscript is well written and experimental data are clearly presented to demonstrate the magnetoelastic coupling.

1. However, it is unclear whether the authors developed and installed new hardware for LA-Ltz-4D-STEM or simply found a customized lens configuration to shorten the camera length using existing lenses. Since the sample studied was amorphous and the data analysis was

relatively insensitive to diffraction conditions, it is uncertain if this technique can be applied to other ferroics in crystalline phases.

Reply:

Thank you for your positive feedback on our manuscript and for emphasizing the key findings of our study. We appreciate your recognition of the development and application of our new method, LA-Ltz-4D-STEM, as well as the clarity of our experimental data. We have revised the manuscript to clarify that we did not modify the hardware of the Themis Z but instead developed customized configurations of lens currents to create a new optical setup for the measurements.

While our initial study focused on an amorphous Fe-based alloy, we believe that the LA-Ltz-4D-STEM technique can indeed be applied to crystalline magnetic and ferroic materials. The core principle of this technique—simultaneously acquiring magnetization and material's structure maps accessed through the large angle diffraction pattern in field-free STEM mode—is broadly applicable. For example, Figure R4b shows a crystalline diffraction pattern recorded in LA-Ltz-4D-STEM mode, demonstrating that, without interference from dynamic scattering, the technique works for thin crystalline samples. We acknowledge that thicker crystalline samples may face challenges due to dynamic scattering, but this can be mitigated through edge detection and electron beam precession. In support of this, we demonstrate the technique in our recent work. Figure R4a-d shows 4D-STEM DPC results for a BTO (BaTiO₃) sample with a thickness of ~ 200 nm, using edge detection and electron beam precession (Figure R4c) and mass-center analysis without precession (Figure R4d).

The dynamic effect is reduced by edge detection and electron beam precession. To validate the method, we studied core charge variations at grain boundaries in crystalline ferroelectrics using an in-situ STM holder and electron beam precession from the Nanomegas system. This approach reduced dynamic effects and diffraction contrast in the DPC measurements, as shown in Figure R4.

Electric field images were generated using the shift of the direct beam, measured via both the center-of-mass and edge-fitting methods for comparison. The variations in core charge at grain boundaries demonstrated the successful separation of the field signal from dynamic scattering in polycrystalline materials. Due to the specific relevance of this work for material applications, we plan to write a separate paper on these findings.

Furthermore, for the diffracted beams, we learned from *Padgett et al., Ultramicroscopy 214 (2020) 112994* that the dynamic effect and diffraction contrast can be suppressed in strain mapping of crystalline samples using the cepstrum method. Our recent work shown in Figure

R5 below confirms that the crystalline structure-induced artifact can be eliminated in 4D-STEM strain mapping through the processing strategy.

Nonetheless, we have included a discussion in the revised manuscript that addresses the potential challenges of studying crystalline materials and highlights the technique's versatility.

Figure R4. 4D-STEM electric field measurement of core charge at a grain boundary in crystalline ferroelectrics using edge detection algorithm and electron beam precession. **a** HAADF image of Fe-doped crystalline barium titanate (BTO). **b** Fitting of diffraction patterns obtained from electron beam precession using edge detection algorithms. 4D-STEM DPC results for a large field of view across multiple grains using **c** edge detection and electron beam precession, and **d** mass-center algorithm without electron beam precession. **f** HAADF image at a grain boundary. **(g)** 4D-STEM DPC results across a single-grain boundary. An external electric bias is applied to modulate space charges at the grain boundary using an in-situ STEM holder.

Figure R5. Strain map of a surface modified SiC, the sample is more than 200 nm thick for keeping residual strain unreleased. **a** virtual bright field image. **b** Averaged diffraction pattern, the red disk highlights the virtual bright detector. **c** ϵ_{xx} map obtained using the cross-correlation method to register the diffraction peaks, the strong variation is due to the variation of

Bragg condition and dynamic scattering. **d** ϵ_{xx} obtained using the cepstrum method. The artifacts that appeared in **c** are eliminated, and the true strain information caused by the surface modification is revealed.

2. Additionally, while the authors claim that atomic structures correlating with magnetic structures can be studied, they only provide strain information. Furthermore, the phenomenon of strain-induced magnetic anisotropy is relatively well known (is this new for this magnetic material?) and not particularly surprising. I think that the authors should justify their findings as a significant advancement to warrant publication in *Nature Communications*.

Reply: We apologize for any confusion regarding this point. While our primary focus was on the correlation between strain and magnetic structures, we acknowledge the importance of including detailed atomic structure information. In the revised manuscript, we provide additional data and analysis demonstrating that LA-Ltz-4D-STEM data enables pair distribution function (PDF) analysis and allows to map the PDF as an atomic structure descriptor of glasses as shown in Figure R6.

Figure R6. Detailed PDF analysis and map of the edge-shared tetrahedra. **a** PDF line scan following the direction of the arrow in Figure 2e perpendicularly to the shear band. The horizontal axis represents the atomic pair distance r (Å) and the vertical axis corresponds to the scan position on the probed material. The temperature-type color corresponds to the amplitude of the PDFs, that is, the population of pair correlation. **b** PDFs from the left (strained) and right (compressed) shear band affected zone (SBAZ). **c** and **d** are the enlarged parts of the first peak and second peak respectively. **e** Map of face-sharing (FS) connected tetrahedra based on the medium-range order (MRO) analysis taking the intensity of PDFs of the STEM-PDF map at $r = 8/3R_1$, R_1 is the averaged first peak position of each PDF, referring to the

method described in [Adv. Mater. 2021, 2007267]. f Schematic of the basic structural types of polyhedra connected by one shared atom (point sharing) and polyhedra connected by two shared atoms (edge sharing).

The temperature-type color corresponds to the amplitude of the PDFs, while the horizontal and vertical axes represent the atomic distance r and the scanning positions on the sample. Shifting features are obvious (in particular at $r \approx 3.4$ and ≈ 4.8 Å) in the SB region, which divides the PDF line scan into two regions: left and right shear band affected zone (SBAZ). For example, two PDFs from left (strained) and right (compressed) SBAZ are clearly distinguished by their first and second peaks. The difference between PDFs indicates the different atomic arrangements in these zones affected by their strains. Figure R6e shows a map of face-sharing (FS) tetrahedra generated from the STEM-PDF map following the approach described in *Advanced Materials* 2021, 2007267. The SB in Figure R6e shows darker intensity differing from its two side neighbors and indicating a reduced population of the FS tetrahedra in SB. The ePDF analysis includes atomic scale information and quantitative analysis to support our claim.

While strain-induced magnetic anisotropy has been explored in many experimental studies, its effects on nanoscale magnetism remain poorly understood due to challenges in correlating magnetic fields with local strain.

The magnetic fields originating from the magnetization of the sample can be measured quantitatively using Lorentz S/TEM (fresnel contrast, holography, DPC, etc), however, strain experiments have typically been designed for simple strain conditions, where strain information is averaged across the entire sample over a micrometer-scale field. Recent studies, such as those in Figures R7 A (*Nature Communications* 2023, 14:3963) and B (*Nature Nanotechnology* 2015, 10(7):589-92), applied uniaxial strain to TEM lamellae but were unable to provide quantitative strain fields at each sampling pixel. Studying magnetic-strain correlations in more complex strain environments, particularly at the nanometer scale, has not been possible.

Additionally, without simultaneous strain imaging, it remains unclear whether local magnetic fields are driven by strain-induced anisotropy or by minimizing magnetostatic energy, making it difficult to study the competition between magnetoelastic and magnetostatic effects.

This issue is even more challenging in metallic glasses with deformation-induced changes to magnetic properties due to (1) imaging the shear bands, which was only recently solved using the strain mapping methods reported in *Advanced Materials* 2023, 35(25): 2212086 and *Advanced Materials* 2021, 2007267, and (2) the complex strain fields that cannot be estimated without direct nanoscale strain imaging. For example, the recent study on the magnetic properties of shear band regions in metallic glass (Figure R7 C, *Nature Communications* 2018, 9:4414) used magnetic force microscopy (MFM) to image the magnetic field at the surface of

the shear band region with sub-micrometer resolution, but provided no structural or strain information, as Lorentz S/TEM is also limited in this regard. The relationship between magnetization and complex local strain fields, particularly how it deviates from low-energy configurations at magnetic or strain domain boundaries, has not been fully understood. Here, the new method, for the first time, visualizes the nanoscale coupling between strain and magnetic fields and demonstrates the competition between magnetoelastic and magnetostatic energy in complex-strained materials, specifically in the shear band region.

Figure R7. **A:** Holographic Lorentz TEM measurement of magnetostriction in a Ni nanostructure during tensile force, cited from Kong et al., *Nature Communications* (2023) 14:3963. **B:** Anisotropic deformation of skyrmions under uniaxial strain, cited from Shibata et al., *Nature Nanotechnology*, 2015, 10(7):589-92. **C:** Surface magnetic domains imaged at a shear-band of a ferromagnetic metallic glass by MFM, scale bar is 5 μm , cited from Shen et al., *Nature Communications*, 2018, 9:4414.

3. It is unclear whether the authors developed and installed new hardware for LA-Ltz-4D-STEM or simply found a customized lens configuration to shorten the camera length using existing lenses. If the former, I would like to recommend that the authors elaborate more in detail about the new lens configuration and installation. In either case, are there any changes in magnetic fields on the sample plane?

Reply: The authors sincerely thank for the reviewer's constructive suggestions. In this work, our development is a customized lens configuration in commercially available hardware. It utilizes the existing Lorentz lens (or the 1st transfer lens in the image corrector), which has never been used in field-free STEM mode, to obtain large-angle diffraction information. This results in a new electron optical setup and experimental strategy. The residual magnetic field without the demagnetization process was measured by a TFS engineer using a Hall effect holder during the installation of the TEM and was found to be between 3-5 mT. This translates to an in-plane magnetic field of 0.5-0.86 mT when the sample is tilted by 10 degrees. We also observed that the field became undetectable after one demagnetization cycle during that

measurement, and we applied this process in the current work. We revised the manuscript to elaborate more in detail about the new lens configuration.

4. It is interesting to measure the phase shift of the electron beam using the center of the 1st amorphous Bragg ring. Quantifying the center-of-mass (CoM) shift of a transmitted and diffracted beam from crystalline samples using a nanoscale electron probe illuminating multiple atomic columns is often complicated due to the dynamic scattering process. If the authors provide full details on CBED pattern data processing such as CoM shift, 1st Bragg ring detection, measuring ellipticity, etc., it will be really useful for people who are interested in this method.

Reply: We follow the review's comment to acknowledge the challenge in crystalline materials due to the dynamic scattering process in the manuscript and also discuss the solutions. The details have been written in our previous answer to Q1. Following the suggestion of the reviewer, we also provide full details on data processing of the 1st Bragg ring detection and ellipticity measurement in the revised manuscript.

5. In the Figure 2a (STEM ADF image) shows a curved shape feature in the sample. Similar curved features are also can be seen in the EFTEM image shown in Figure S3. It seems that the relative density map (Figure 2d) shows that reduced density can be seen near the sample edge and between two pop-in/outs. The strain changes across the pop-in and pop-out shear bands also appear to be suppressed in the reduced density area. Can authors more elaborate on this?

Reply: The curved feature arises from thickness variations during the preparation of the FIB-TEM lamella.

The area exhibiting a reduced density in Figure 2d in the manuscript is about 300 nm thick. The apparent density reduction is presumably due to inelastic scattering that happened in the thick sample, which results in a smooth background that is higher in low angles and gradually increases for larger angles. Adding this background shifts the peak position of the 1st diffracted ring slightly resulting in an underestimation of the density. The reduced SNR due to the strong background may reduce the sensitivity for detecting the diffraction ring and reduce the contrast of the shear band, blurring alternations. This may also affect the strain measurement in the thick area. It may be expected that using an energy-filtered 4D-STEM can improve the situation. For thinner samples, such as the 200 nm thick sample, the data processing procedure works well and the shear band features are more clearly visible. We appreciate the reviewer for pointing this out. We added this discussion in the revised manuscript for clarification.

[figure redacted]

Figure R8: Strain observation for the sample at different thicknesses prepared from the deformed Fe-based metallic glass ribbon. [Ultramicroscopy 255, 113844 2024] **a** Volumetric strain map of the sample that was thinned in several steps from (top to bottom) 300 nm to 200 nm, 130 nm, and 70 nm. The same area was mapped by 4D-STEM using identical imaging settings. **b** Line profiles of the volumetric strain maps across a shear band for the sample with different thicknesses.

6. In Figures 2, 3, and 4, most important data showing the magnetoelastic coupling, there are clear differences in magnetic domain structures between the pop-in and pop-out regions. In the pop-out region, compressive strain perpendicular to the shear band direction and tensile strain parallel to the shear band direction are observed. This trend seems to be easily understood by the conventional Poisson effect. In this respect, even though the pop-in region has a simple 90° rotation strain field relationship with respect to the pop-out one, why does the magnetic domain have much larger magnetic domains than those in the pop-out region?

Reply: For magnetically soft ferromagnetic materials, the physical size of the domains is typically large to minimize the density of domain walls which are higher energy compared to domains. As this material has a positive saturation magnetostriction (~40 ppm), the magnetoelastic energy tends to align magnetic moments parallel to the tensile strain direction. Strain-induced anisotropy in the pop-out side is along the SB direction, which is the geometric direction (i.e. the long edge) of the strain zone. The geometric anisotropy of the magnetization (resulting from the minimization of the magnetic static energy) and the strain-induced magnetic anisotropy (resulting from the minimization of magnetoelastic energy) are in line with each other on the pop-out side. In contrast, the strain-induced magnetic anisotropy in the pop-in side falls in the geometrically short direction of the strained zone. So that the geometric anisotropy and strain-induced anisotropy are perpendicular to each other on the pop-in side. To minimize both the magnetostatic and magnetoelastic energy terms, spins oriented parallel to the SB with a large single domain form on the pop-out side, whereas smaller domains and a number of domain walls form on the pop-in side. Similar magnetic domain structures have been observed in samples with orthogonal magnetic anisotropies in soft magnetic amorphous thin films induced by ion irradiation and locally induced stress variation [*Applied Physics Letters* 2009, 94(6)]. We have highlighted this explanation in red in our revised manuscript.

To Reviewer #4:

1. The authors report a study of mapping magnetic spin textures and strain field of a Fe-based amorphous alloy simultaneously by using 4D-STEM in a Lorentz mode. By evaluating

the center shift and distortion (ellipsoid shape) of the first diffraction ring, the magnetic induction, strain and atomic relative density maps can be retrieved. Along with the statistical analyses, the authors present the coupling between magnetoelastic and magnetostatic energies in this material. The field-driven behavior was also explored and the results show that the deformed alloy has low hysteretic behavior. This work is of interest and technically sound, and the paper is well-written.

Reply: Thank you for your positive feedback on our manuscript and for highlighting the key findings of our study.

2. However, the measurement of magnetic induction based on DPC method has been well developed. The approach to estimate the strain field etc. have already been reported by other work Ref [29] and Authors' other publications Refs [23] [31]. The authors mention they have used the first transfer lens of the imaging Cs-corrector as the Lorentz lens for controlling the diffraction space information. This overall limits the applicability of the approach as the microscope needs a Cs-corrector and therefore cannot be applied to any commercial microscope. Using mini-objective lens or Lorentz lens including use of Cs-corrector's first transfer lens also has been previously documented.

Reply: The key development of the reported method is that it enables simultaneous mapping of both magnetic and structural information, whereas DPC in field-free mode with a conventional microscope setup can only provide the magnetic information. As described in the manuscript, the reported method can be implemented either using the first transfer lens of an imaging Cs-corrector or a Lorentz lens, and it is thus not strictly limited to Cs-corrected TEM. Many modern TEMs include a Lorentz lens or a Cs corrector enabling widespread application accessible in the field. So far, Lorentz lenses have only been used for field-free TEM mode by manufacturers or research groups, but our new configuration uses them in the STEM mode for the first time to achieve the new possibility.

3. Furthermore, the authors claim of correlating magnetic structure with atomic structure is a little overstated. At best, the current work can be considered correlating microstructure with magnetic domain structure. There isn't really atomic scale information presented or derived from the dataset. The insights derived into the correlation of magnetic domains and strain has been studied extensively using Fresnel mode LTEM. The authors work does not provide any novel insights into the coupling. Based on these remarks, the work/approach does not meet the required degree of novelty of significance in the field to be published in Nature Communications. Nevertheless, the Authors do provide a useful method to gain insight into the correlation between strain and magnetic domains, which would be potentially interesting

for specific communities. Therefore, this work should be submitted to a more specialized journal.

Reply: Our study presents a novel application of LA-Ltz-4D-STEM, enabling simultaneous nanoscale mapping of magnetization and strain.

We acknowledge that the atomic structure was not directly imaged, but the diffraction pattern with the high-angle signal enabled by our new electron optical setup allows to represent the atomic arrangement, i.e. the distance between atoms and their orientation. This atomic structure information was used to calculate the strain and density maps. Electron-based pair distribution function (ePDF) analysis can be performed simultaneously with the magnetic and strain maps as shown in Figure R6. ePDF describes the atomic short- and medium-range order as the most effective structure descriptor for disordered materials (Please refer to the answer of Q2 from Reviewer #2). In the revised manuscript, we provide the ePDF map along with the magnetic and strain images demonstrating the capability of the reported method for simultaneous mapping structure and magnetic information. Moreover, following the strategy in electron ptychography reported in Chen et al., *Nature Nanotechnology* 2022, **17**(11): 1165-1170 and, in particular, the results demonstrated by Humphry et al., *Ptychographic electron microscopy using high-angle dark-field scattering for sub-nanometre resolution imaging*, *Nature Communications* 2012, **3**, 730, the large angle diffraction signals have been shown to enable direct lattice imaging.

While previous studies, including those using Fresnel mode LTEM, have explored the correlation between magnetic domains and overall average straining of samples, the strain-induced effects on magnetism on the nanoscale remain poorly understood due to the experimental challenge of correlating the magnetic fields to the local strain and its nanoscale variation. The magnetic fields originating from the magnetization of the samples' magnetization can be measured using Lorentz S/TEM (fresnel contrast, holography, DPC, etc), however, strain experiments have typically been designed for simple strain conditions, where strain information is averaged across the entire sample over a micrometer-scale field. Recent studies, such as those in Figures R7 A (*Nature Communications* 2023, 14:3963) and B (*Nature Nanotechnology* 2015, 10(7):589-92), applied uniaxial strain to TEM lamellae but were unable to provide quantitative strain fields at each sampling pixel. Consequently, studying magnetic-strain correlations in more complex strain environments, particularly at the nanometer scale, has not been possible.

Additionally, without simultaneous strain imaging, it remains unclear whether local magnetic fields are driven by strain-induced anisotropy or by minimizing magnetostatic energy, making it difficult to study the competition between magnetoelastic and magnetostatic effects.

This issue is even more challenging in metallic glasses with deformation-induced changes to magnetic properties due to (1) imaging the shear bands, which was only recently solved using the strain mapping methods reported in *Advanced Materials* 2023, 35(25): 2212086 and *Advanced Materials* 2021, 2007267, and (2) the complex strain fields that cannot be estimated without direct nanoscale strain imaging. For example, the recent study on the magnetic properties of shear band regions in metallic glass (Figure R7 C, *Nature Communications* 2018, 9:4414) used magnetic force microscopy (MFM) to image the magnetic field at the surface of the shear band region with sub-micrometer resolution, but provided no structural or strain information, as Lorentz S/TEM is also limited in this regard. The relationship between magnetization and complex local strain fields, particularly how it deviates from low-energy configurations at magnetic or strain domain boundaries, has not been fully understood. Here, the new method, for the first time, visualizes the nanoscale coupling between strain and magnetic fields and demonstrates the competition between magnetoelastic and magnetostatic energy in complex-strained materials, specifically in the shear band region.

Figure R7. **A:** Holographic Lorentz TEM measurement of magnetostriction in a Ni nanostructure during tensile force, cited from Kong et al., *Nature Communications* 2023, 14:3963. **B:** Anisotropic deformation of skyrmions under uniaxial strain, cited from Shibata et al., *Nature Nanotechnology* 2015, 10(7):589-92. **C:** Surface magnetic domains imaged at a shear-band of a ferromagnetic metallic glass by MFM, scale bar is 5 μm , cited from Shen et al., *Nature Communications* 2018, 9:4414.

4. in the introduction, the statement, “The magnetic structure is intrinsically determined by the interatomic distances and coordination.”, is not entirely correct. The magnetic structure is also strongly dependent on the chemical elements. Also, under the circumstances that are presented in the authors work, the more stronger contribution to the magnetic structure will be from shape anisotropy (magnetostatic energy) due to the thin TEM lamella.

Reply: We thank the reviewer for pointing out the potential misleading of the sentence. We have revised the sentence to make it clearer.

Additionally, in the context of our work involving thin TEM lamellae, shape anisotropy (magnetostatic energy) can indeed play a significant role. This holds universally for all magnetic measurements in TEM. We appreciate this feedback and revised the introduction to more accurately reflect these factors.

5. What materials is the diffraction patterns taken from in the Figure 1b top? The Author could label clearly the materials for Figure 1b in the caption as I thought both diffraction patterns are from the Fe_{85.2}Si_{0.5}B_{9.5}P₄Cu_{0.8}. Figure 1b shows the diffraction patterns of crystal with multi-phases and list the pair distribution function, which however are not very relevant to current work based on the discussion in the paper and causes confusion. This information could be included in the supplementary or could just cite relevant paper. This is just a minor suggestion.

Why do the authors present the diffraction ring of SmCo not that of Fe-based alloy in Figure 1b bottom?

Reply: Thank the reviewer for pointing out the missing information in the figure caption. The diffraction ring in Figure 1b is from a Fe-based alloy. The crystalline pattern is from the SmCo permanent magnet to demonstrate the method is applicable to crystalline materials. The list of ePDF originally aims to show it can be one of the atomic structure analyses. We added an exemplary ePDF analysis in the revised manuscript. We also follow the reviewer's suggestion to add more description to the caption.

6. Were there additional configurational changes (access to lens currents, vendor specific hardware access) that were needed to achieve the higher diffraction angle in the setup presented?

Reply:

The diffraction, intermediate, and projection lenses were also adjusted to obtain the clear transform of the diffraction pattern and set the desired camera length. Their currents are camera length dependent and optimally calculated according to the demands. For the case with an imaging corrector, additional tuning of the multipoles inside the corrector is required to minimize the diffraction distortion.

7. Could author make red lines (representing compressive field) in Figure 2e more obvious, such as, increasing line thickness? It is hard to see them from a colorful background.

Reply: For clarification, we have separately mapped both arrow vectors in a white background (Figure R9) and included it in the supplementary material.

Figure R9: Visualization of the magnetic field and atomic strain coupling using LA-Ltz-4D-STEM observation of a plastically deformed amorphous metallic alloy. **a** The local \vec{B} field is represented by white arrows and the local $\vec{\epsilon}_{\text{ten}}$ field is represented with red sticks. **b** The local \vec{B} field is represented by white arrows and the local $\vec{\epsilon}_{\text{com}}$ field is represented with red sticks.

8. How does the relative density that the authors have calculated compare with the HAADF image intensity?

Reply: The HAADF image intensity shows a strong correlation with the EFTEM thickness map (Figure S3), whereas the density variations are not as clearly visible in the HAADF-STEM images as the HAADF mixes the density and thickness variation of the lamella. The thickness variation is much stronger than the density variations. More details have been discussed in the answer to question Q5 from reviewer #3. In our study, the relative atomic density can be determined by quantifying the diffraction ring. It effectively separates density information from thickness variations.

9. The authors mention they also performed imaging using conventional 4D-STEM imaging. If there is a strong magneto elastic coupling, introducing a relatively strong field and out of plane magnetization will affect the local strain and shear bands. How did the authors account for this?

Reply: The authors sincerely appreciate the interesting comment. Indeed, we also questioned the Z-axis (thickness direction) dependency of the deformation state and magnetization during our work. One challenging aspect is that TEM only provides information about the projected sample structure, leading to the intrinsic limitation that signals are inevitably averaged along the thickness direction.

We conducted a conventional 4D-STEM measurement (in a fully out-of-plane magnetized state due to the strong magnetic field generated from the objective lens) shown in Figure S6 in the supporting information intending to rule out any artifacts in the magnetic and strain image. In the DPC image of the fully z-axis magnetized state (Figure S6b), none of the contrast variations across the shear bands were visible, confirming that our observation of the magnetic field originates intrinsically from the sample's magnetic structure. However, we did observe slight changes in the amplitude of the strain field within the sample between Lorentz and fully saturated conditions. As the same as the reviewer's expectation, we believe this is due to the magnetostrictive effect through magnetoelastic coupling as the reviewer pointed out. Nonetheless, the slight variation of residual strain field ($<0.05\%$) shown in Figure S6g is much smaller than the shear banding-induced strain, allowing us to maintain the same conclusion.

10. The authors state that the magnetic moments are aligned parallel to the tensile strain direction. These results are not unexpected. Also such effects have already been demonstrated using Fresnel LTEM imaging where changes in domain wall and domain orientations can be easily correlated with diffraction patterns and therefore strain/crystallographic directions.

Reply: Following the reviewer, we revised our manuscript to provide more information on previous works, e.g. Fresnel mode LTEM, in the introduction and provide a comparison to the advances with our new method. We have partially addressed this in the previous reply at the beginning of the answers for Q3. While previous studies, including those using Fresnel mode LTEM, have explored the correlation between magnetic domains and strain, our method provides a new level of resolution and local details for the correlative analysis in contrast to the LTEM images correlated with a few diffraction patterns averaged over the sample area. Using the new method, we observed the competition of the strain-anisotropy and magnetostatic energy as well as the domain wall energy in the sample, beyond the simple concept of the alignment between tensile strain and magnetic moment. The asymmetric strain field across the shear band leads to the uncommon domain structure on both sides of the shear bands, in particular in the pop-in side, the magnetization is forced to the geometrically high energy direction, and the region is split to multiple small domains through the energetic

competition. At the shear band vicinity, magnetic moments break the common expectation of “*the magnetic moments are aligned parallel to the tensile strain direction*”. A significant part of them were observed against the strain-induced anisotropy and form closure domains (Figure 3f and h). Furthermore, to distinguish the magnetic moments following or against the strain field that are intricate at nanoscale, e.g. the analysis of Figure 3f-h in the main manuscript, is only possible using the new method. It is the first time to see and explain the influence of nanoscale strain variation on magnetic moments in metallic glass.

Moreover, the correlation between LTEM image and a few diffraction patterns still requires switching the microscope between the field-free mode and standard field-on mode. This not only hinders to check of the magnetic state after a diffraction experiment but also destroys the in-situ experimental possibility. The newly developed method can solve this issue.

11. The authors state that the closure domains are formed to reduce magnetic anisotropy. This statement is incorrect, and the authors should be careful about this. Closure domains would be a results from the shape anisotropy rather than magnetic anisotropy.

Reply:

We thank the reviewer for pointing out our unprecise expression. We have revised the manuscript to clarify that closure domains form to minimize the magnetostatic energy.

12. Overall in the analysis of strain, there was no error bar stated. These measurements are based on slight changes in the distortion of the diffraction ring, and therefore can be very sensitive to the measurement, intensity, area on the detector. An associated error bar in the measurements should be included. Also what is the sensitivity level for the measurement of strain and magnetic induction map. With the higher diffraction angle, as the authors mention, the measurement of deflection is effectively smaller in the diffraction plane. Can the authors provide some comparison and how statistically significant the measurements are?

Reply: Following the suggestion, we calculated the standard deviation of the strain field (volumetric strain as an example) using the data from the undeformed sample to be 0.08%. This is an upper limit for the accuracy of the measurement as it is also affected by local structural variations, which intrinsically exist in the metallic glasses. Nevertheless, the small standard deviation is well below the shear band features in our experimental observation.

We measured the standard deviation of magnetic variation within a magnetic domain of the ferromagnetic amorphous alloy used in this study, recording values as low as 10 mT for the direct beam and 70 mT for the first ring. These values are considerably smaller than the overall magnetization (~1 T) within the domain. To prevent damage to the camera from the direct beam, a beam stopper was used, and the first ring was utilized for the experiment.

13. *Considering that Figure 3 shows statistical analysis of the coupling between magnetic field and tensile field, it would be very helpful to understand the results by replacing the compressive field map (the red lines in Figure 3(a)) with tensile field map to avoid confusion.*

Reply: We completely understand the reviewer's point. As the positive saturation magnetostriction ($\lambda_s \sim 40$ ppm) of the Fe-based metallic glass, the e_{ME} tends to align magnetic moments parallel to the tensile strain direction. We intentionally use the tensile field map for the statistical analysis as the tensile field induces a major easy axis for this material. We have tried to show the strain field in Figure 3a using the tensile strain rather than the compressive strain. However, the red lines overlapped with the magnetic vectors (the white arrows) and made the figure very unclear for visualization as shown in Figure R9.

14. *The author should mark the external in-plane field value in Figure 4(a-f), which may allow to easily understand the figure. In addition, I would suggest to make it clear that the Figure 4g and h were recorded at zero field, as the Figure h seems showing the spin textures under an external field with the label of "After magnetization".*

Reply: We followed the reviewer's suggestions and revised the manuscript accordingly.

15. *The nearly identical spin textures was achieved after removing magnetic field in the deformed sample. How about the undeformed sample? Have the authors seen the vortex restored when reducing the field strength to zero?*

Reply: Regarding the undeformed sample, we have observed a similar trend where the vortex-like spin textures at the center of the lamella in Figure S5 and Supplementary Video 2 were restored when the external magnetic field was reduced to zero. It is worth noting that the undeformed sample may retain residual stress anisotropy from fast melt-spin quenching and FIB thinning, although it is very low and nanoscale homogeneous compared to the situation of the deformed sample, contributing to variations in the magnetic domain structure compared to regular Landau-like domain structures. These variations are also evident in the curvature of the domain walls and their deviation from a standard closure domain angle.

16. *There is the lack of the discussion of strain-induced magnetic properties in the abstract, introduction or conclusion.*

Reply: We follow the reviewer's suggestion to add a more prominent discussion of strain-induced magnetic properties in the abstract, introduction, or conclusion sections of the man-

uscript. In the revised manuscript, we include a more concise statement to highlight the exploration of strain-induced magnetic properties in the studied samples. We expanded the introduction to outline the motivation behind studying strain effects on magnetic properties and how our experimental approach contributes to this understanding. We emphasize the implications of our findings on strain-induced magnetic properties and their relevance to understanding material behavior in the conclusion section.